# Association between Diet and Xerostomia: Is Xerostomia a Barrier to a Healthy Eating Pattern?

**DOI:** 10.3390/nu13124235

**Published:** 2021-11-25

**Authors:** Indre Stankeviciene, Jolanta Aleksejuniene, Alina Puriene, Lina Stangvaltaite-Mouhat

**Affiliations:** 1Institute of Dentistry, Faculty of Medicine, Vilnius University, 01513 Vilnius, Lithuania; indre.stankeviciene@mf.vu.lt (I.S.); alina.puriene@mf.vu.lt (A.P.); 2Department of Oral Health Sciences, Faculty of Dentistry, The University of British Columbia, Vancouver, V6T BC, Canada; jolanta@dentistry.ubc.ca; 3Lina Stangvaltaite-Mouhat, Department of Clinical Dentistry, Faculty of Health Sciences, UiT The Arctic University of Norway, 9037 Tromsø, Norway; 4Oral Health Centre of Expertise in Eastern Norway, 0369 Oslo, Norway

**Keywords:** xerostomia, autoimmune diseases, diet, adults

## Abstract

Objective. Xerostomia is a subjective feeling of dry mouth and is commonly observed in patients with autoimmune diseases. Our study examines the association between xerostomia and diet. Materials and Methods. The cross-sectional study includes 1405 adults from 15 Lithuanian geographical areas (52% response rate). A self-reported questionnaire inquired about xerostomia, sex, age, education, residence, and consumption of selected 23 diet items. For the multivariable analysis, 23 diet items were categorized into eight major diet groups. The data were analyzed by bivariate and multivariable analyses. Results. When comparing participants with and without xerostomia, there were significant differences in consumption frequencies concerning cold-pressed oil (*p* = 0.013), bread (*p* = 0.029), processed meat products (*p* = 0.016), fat and lean fish (*p* = 0.009), and probiotic supplements (*p* = 0.002). In the multivariable binary logistic regression model, when controlled for other determinants, the higher consumption of carbohydrates (OR 0.39, 95% CI 0.23–0.65), proteins (OR 0.56, 95% CI 0.32–0.99), and oils (OR 0.58, 95% CI 0.34–1.00) was associated with a lower likelihood of xerostomia. Conclusions. The association between xerostomia and the consumption of the six diet items—cold-pressed oils, lean and fat fish, bread, processed meat, and probiotic supplements— and the three major diet groups—carbohydrates, proteins, and oils—was observed. Longitudinal studies are needed to validate the observed associations.

## 1. Introduction

Xerostomia is defined as a subjective symptom of dry mouth, which may affect swallowing, chewing, taste perception and is the most important predictor in reduced oral health-related quality of life; also, xerostomia has been associated with nutritional changes [1,2,3]. Xerostomia is a common condition of patients with autoimmune diseases; 50% or more patients with systemic lupus erythematosus, polymyositis/dermatomyositis or systemic scleroderma reported xerostomia [4]. Sjogren‘s syndrome is one of the three most prevalent systemic autoimmune diseases and 98% of patients reported to have xerostomia [5]. Although xerostomia is often claimed to lead to an impaired nutrition, there is lacking knowledge of which specific food groups associate with this condition. Therefore, our study examines the association between xerostomia and 23 diet items.

## 2. Materials and Methods

### 2.1. Study Design and Participants

The cross-sectional study analyzed data collected during the Lithuanian National Oral Health Survey. The data were collected in 2017–2019, and the study included 1405 (52% response rate). The details about the sample size calculation and participant recruitment are presented elsewhere [6,7]. The selected number of participants (based on the calculation of minimum sample size) was randomly selected from the patient lists at primary health care institutions in five largest cities and randomly selected peri-urban and rural areas in each of the 10 Lithuanian counties. The inclusion criteria were subjects aged 35–74 years (based on the previous National Oral Health Survey and WHO recommendations).

### 2.2. Questionnaire

Participants were asked to complete the WHO Oral Health Questionnaire for Adults, the items for which were specified according to the country-relevant context [8]. The questionnaire was translated from English to a Lithuanian language, and then back translated to English by two independent persons with all inconsistencies being discussed later, and the same process was performed for two other minority languages: Russian and Polish. To cover food groups, included in a healthy food pyramid, the original WHO questionnaire, which presented 8 diet items, was supplemented with 12 diet items as well as with 3 types of supplements, namely, omega 3, fish oil, and probiotic supplements. The whole questionnaire was pretested in 10 adults who were not included in the main study. 

For all diet items, the participants were asked to indicate their frequency of consumption as follows: ‘1’—seldom/never; ‘2’—several times a month; ‘3’—once a week; ‘4’—several times a week; ‘5’—every day; ‘6’—several times a day. For statistical analyses, responses regarding the diet items, i.e., use of vegetables, fruits, cold-pressed oils, linseed oil, bread, meat, processed meat products, oily fish, lean fish, olive oil, refined oils, rice and pasta, cereals, potatoes, eggs, dairy, fermented dairy, nuts and seeds, fish oil supplements, probiotic supplements, and omega 3 supplements were grouped into the following categories: ‘1’—product used seldom/never or several times a month; ‘2’—once a week; ‘3’—every day or several times a day. The variable “use of sweetened drinks” was computed summing up four categories of drinks: juice, tea with sugar, coffee with sugar, and soft drinks with possible scores of ‘1’—seldom/never; ‘2’—several times a month; ‘3’—once a week; ‘4’—several times a week; ‘5’—every day; ‘6’—several times a day; subsequently, the sum score was categorized based on quartiles into ‘1’—low frequency; ‘2’—medium frequency; ‘3’—high frequency. Similarly, the variable “consumption of desserts with added sugar” was composed by summing up the scores regarding the use of these products: (1) biscuits, cakes, cream cakes, (2) sweet pies, buns, (3) candies, chocolate (Table 1). In preparation for multivariable binary logistic regression analysis, subjects were grouped into lower and higher specific food/drink consumption groups (based on median value) concerning eight major diet groups (proteins, carbohydrates, oils, fruits and vegetables, dairy, supplements, sweetened drinks, and desserts with added sugar). The variable age (in full years) was dichotomized into two age categories (for females to reflect the menopausal time) (<55 years) and (55 years or older) [9]. This way, we prepared 10 predictors consisting of 2 sociodemographic determinants (sex and age) and 8 major diet groups as diet-related determinants.

To assess xerostomia, participants were asked “How often does your mouth feel dry” with possible answers ‘never’, ‘sometimes’, ‘often’, and ‘always’ [10]. The xerostomia group included those who answered ‘often’ or ‘always’ to the question “How often does your mouth feel dry”, and the non-xerostomia group included those who answered ‘never’ or ‘sometimes’.

### 2.3. Statistical Analyses

The statistical analyses were performed with the SPSS version 26 (IBM SPSS, Armonk, NY, USA). The difference in distribution of four socio-demographic characteristics (age, sex, education, and residence) and frequency of consumption (low, medium, and high) of 23 diet items between xerostomia and non-xerostomia groups were tested using Chi-square test for categorical variables and the Mann–Whitney U test for continuous variables. 

Multivariable binary logistic regression analysis was used to determine the association between xerostomia, frequency of consumption of eight major diet groups (higher and lower) and two socio-demographic characteristics. For selection of determinants for the multivariable binary logistic regression model, it is commonly recommended to have a minimum of 10 subjects for the events category (here, the presence of xerostomia) [11]. In our study, there were 112 subjects with xerostomia; consequently, we could not include more than 11 determinants. This requirement was fulfilled as we tested a total of 10 determinants, including two socio-demographic characteristics (female sex and older age) and major diet groups. The testing showed the absence of multicollinearity as indicated by low Variance Inflation values (<1.5); therefore, we considered findings from the binary logistic regression model with the backward LR selection of predictors to be valid. The final model emerged in seven steps/iterations and selected four significant predictors. The statistical significance for all tests was set at *p* < 0.050. Crude (for all tested determinants) and adjusted odds ratios (ORs) (for the determinants of the final model) with their 95% confidence intervals (CI) were calculated [12,13].

## 3. Results

Out of 1405 participants, the majority was female (66.9%), residing in urban areas (71.6%), and the mean (sd) age of participants was 55 years (11.9). Of all, 76.7% (N = 1077) never experienced dry mouth, 15.4% (N = 216) experienced it sometimes, 5.8% (N = 81) often, and 2.2% (N = 31) always. The participants reporting xerostomia often or always composed the xerostomia group (in total 8%, N = 112), and the others were allocated into the non-xerostomia group. The higher proportion of participants reporting xerostomia were female, of older age, and had less years of education (Table 2).

Overall, there was a considerable variation among the study participants with xerostomia concerning the consumption of 23 diet items (Table 3). When comparing participants in the non-xerostomia group with those reporting xerostomia, there was a statistically significant difference in the consumption frequency of cold-pressed oil, bread, processed meat products, fat fish, lean fish, and probiotic supplements (Table 3).

According to the multivariable binary logistic regression model, when controlled for other determinants, the higher likelihood of xerostomia was associated with persons 55 years old or of older age (OR 1.66, 95% CI 1.03–2.69) (Table 4). The lower likelihood of xerostomia was associated with a higher consumption of carbohydrates (OR 0.39, 95% CI 0.23–0.65), proteins (OR 0.56, 95% CI 0.32–0.99), and oils (OR 0.58, 95% CI 0.34–1.00).

## 4. Discussion

Recently, the topic of nutrition has been brought to the center of the attention as being related to the risk of autoimmune diseases and their progression. It has been reported that nutrition has a potential either to increase the risk for autoimmune diseases or, alternatively, be a protective factor for them [14]. The current study found that a more frequent consumption of carbohydrates, proteins, and oils was associated with a lower likelihood of xerostomia. Additionally, a higher consumption of cold-pressed oils, lean and fat fish, and probiotic supplements, and a lower consumption of bread and processed meat was associated with xerostomia. 

The strength of the present study was a relatively large sample size of 1405 adults, the study sample covering a wide geographical and socioeconomic area in Lithuania, namely, 10 randomly selected rural/peri-urban areas one from each of the 10 counties, in addition to the 5 biggest cities in Lithuania. Moreover, the questionnaire inquired about a variety of diet items, covering main food categories, including those from the healthy food pyramid and those potentially harmful for health. 

The limitation was the cross-sectional study design that did not allow causal inferences, e.g., it could not be defined whether xerostomia was a consequence of a specific diet or the participants reporting xerostomia tended to prefer specific types of food. For that reason, prospective studies are needed to determine the causal relationship. Furthermore, the study questionnaire inquired about the frequency of the consumption of preselected diet items during the past month, this may not be a true reflection of individual dietary patterns. 

Our study found that participants who reported a higher consumption of oils had less likelihood of xerostomia. There was evidence that plant-derived oils containing omega 3 fatty acids, mainly α-linolenic acid, may be valuable in the prevention and treatment of various health disorders [15]. A randomized placebo trial performed with patients suffering from Sjogren’s syndrome demonstrated that both receiving wheat germ oil supplements (as placebo) and a supplement “n-3” including flaxseed oil and vitamin E contributed to an increased salivary flow. The authors discussed that the supplementation of plant-derived oils, presenting omega 3 fatty acids, may be beneficial for patients with autoimmune diseases to improve their dry mouth symptoms [16]. A cross-sectional study reported that patients with Sjogren’s syndrome were deficient in the intake of omega 3 fatty acids [17]. This finding was in line with the results of another xerostomia-related study, where the intake of different fatty acids was tested, but only a deficiency in omega 3 fatty acids was significantly related to xerostomia [2]. On the other hand, in the bivariate analysis, we found that a more frequent use of cold-pressed oils was associated with xerostomia. This may be due to the fact that, in Lithuania, cold-pressed coconut oil is commonly used. A meta-analysis, published in 2020, reported that the use of coconut oil compared to non-tropical vegetable oils was significantly associated with a higher DL cholesterol and could be related with chronic diseases; thus, it may increase the risk of xerostomia [18]. Further research is needed to understand the role of different plant-derived cold-pressed oils and omega 3 fatty acids supplementation in the development and management of xerostomia and autoimmune diseases.

In our study, the lower use of carbohydrates was related to a higher likelihood of xerostomia. In addition to this, a significant bivariate association was found between xerostomia and the low consumption of bread. A similar study found that participants suffering from Sjogren’s syndrome had a lower intake of bread compared to the control group [19]. The authors suggested that this may be due to the fact that starchy products are considered to be dry; thus, experiencing xerostomia patients avoided them. The lower intake of bread among participants with Sjogren’s syndrome resulted in the overall deficiency of the intake of carbohydrates that subceeded below the recommendations [19]. More severe dry mouth symptoms were associated with a lower intake of whole grain products as indicated by the avoidance of certain carbohydrate-containing foods, such as cereal, rice, and pasta [20]. No association between xerostomia and other types of carbohydrates (except bread) was observed in the bivariate analysis. This may be related to cultural food preferences, as pasta and rice is not a part of the traditional Lithuanian cuisine. It is likely that patients having xerostomia tend to avoid using carbohydrate-rich food, which might result in a carbohydrate deficiency. In the case of a severe carbohydrate deficiency, saliva substitutes may benefit, as it is known that they improve the symptoms of dry mouth and increase swallowing ability [3]

In our study, a high-frequency consumption of lean and fat fish was related to xerostomia. This finding was in line with an earlier mentioned study, which found that patients suffering from Sjogren’s syndrome tended to eat more fish than subjects in the control group [19]. The authors discussed that the smooth and viscous texture of fish as compared to meat may be better tolerated by patients with xerostomia, and those suffering from Sjogren’s syndrome and other autoimmune diseases [19]. In support, our study found that a high proportion of participants having xerostomia reported a low consumption of processed meat products. Additionally, a higher likelihood of xerostomia was observed for participants with a lower consumption of proteins. This may be explained by the preference of meat over fish in the traditional Lithuanian cuisine. More research may be warranted to examine the association between specific protein and xerostomia.

In our study, a high-frequency consumption of probiotic supplements was associated with xerostomia. In contrast, another cross-sectional study suggested that probiotics may lower the risk of xerostomia as their use increases salivation [21]. Gut dysbiosis was linked to several autoimmune diseases such as rheumatoid arthritis, systemic lupus erythematosus, Behcet’s disease, and Sjogren’s syndrome [22]. In addition, several studies associated Sjogren’s syndrome with the antibiotics-induced gut dysbiosis [23]. Although we did not obtain a clear explanation, it may be that a higher consumption of probiotic supplements in the xerostomia group was due to the need to take them after an antibiotic treatment or in accordance with other conditions and symptoms related to gut dysbiosis. Therefore, this result of our study should be interpreted with caution and further research in this field is needed.

Overall, the current study showed that there was a substantial variation among our study participants concerning the consumption of 23 selected diet items. Seemingly, experiencing xerostomia did not limit the ability to eat various types of food. However, an avoidance of and preference for some specific food categories may result in an unbalanced diet. Therefore, patients with xerostomia could benefit from health and balanced diet advice as it is important in maintaining good health and preventing other health conditions. Special care should be taken to optimize the intake of high-quality carbohydrates, oils, and proteins. Patients with difficulties in consuming sufficient amounts of proteins could be advised to choose fish as an alternative to meat products, or salivary substitutes should be prescribed for improving the ability to swallow more dry food products. Probiotic supplements may be suggested for patients with xerostomia and gut dysbiosis-related condition, such as autoimmune diseases. In addition, our study did not find a significant relationship between xerostomia and the intake of important food groups, namely, vegetables and fruits, or dairy products. Therefore, in general, xerostomia may be compatible with the ability to maintain a balanced diet important for overall health. 

## 5. Conclusions

An association between xerostomia and the consumption of six diet items—cold-pressed oils, lean and fat fish, bread, processed meat, and probiotic supplements—and three major diet groups—carbohydrates, proteins, and oils—was observed. Longitudinal studies are needed to validate the observed associations.

## Figures and Tables

**Table 1 nutrients-13-04235-t001:** Operationalization of study variables and their categorization for statistical analyses.

Variables	Measurement Original Values	Categorization for Analyses
Socio-Demographic Characteristics		
Sex	Males Females	Males Females
Age	“How old are you?” (in full years)	No categorization
Residence	Urban Peri-urban Rural	Peri-urban/rural Urban
Education	“How many years have you spent in education?”	No categorization
Diet items:	“How often do you eat or drink any of the following foods or drinks, even in small quantities?” (for each diet item)	
Vegetables, fruits, cold-pressed oils, linseed oil, bread, meat, processed meat products, oily fish, lean fish, olive oil, refined oils, Rice and pasta, cereals, potatoes, eggs, dairy, fermented dairy, nuts or seeds, fish oil supplements, probiotic supplements, omega 3 supplements.	Seldom/never Several times a month Once a week Several times a week Every day Several times a day	Seldom/never/several times a month Once a week Every day/several times a day
Sweetened drinks (Total consumption indicated by the sum of all items)	“How often do you eat or drink any of the following foods or drinks, even in small quantities?” (1) Juice, (2) tea with sugar, (3) coffee with sugar, (4) soft drinks:	
	Seldom/never Several times a month Once a week Several times a week Every day Several times a day	Sum of 7 or less Sum of 8–12 Sum of 13 or more Lower consumption (sum of <10) Higher consumption (sum of ≥10)
Desserts with added sugar (Total consumption indicated by the sum of all items)	“How often do you eat or drink any of the following foods or drinks, even in small quantities?” (1) Biscuits, cakes, cream cakes, (2) sweet pies, buns, (3) candies, chocolate:	
	Seldom/never Several times a month Once a week Several times a week Every day Several times a day	Sum of ≤6 Sum of 7–9 Sum of ≥10 Lower consumption (sum of <8) Higher consumption (sum of ≥8)
Carbohydrates (Total consumption indicated by the sum of all items)	“How often do you eat any of the following foods or drinks, even in small quantities?” (1) Potatoes, (2) cereal, (3) pasta or rice, (4) bread:	
	Seldom/never Several times a month Once a week Several times a week Every day Several times a day	Lower consumption (sum of <15) Higher consumption (sum of ≥15)
Proteins (Total consumption indicated by the sum of all items)	“How often do you eat any of the following foods or drinks, even in small quantities?” (1) Meat, (2) processed meat, (3) lean fish, (4) fat fish, (5) eggs, (6) nuts or seeds:	
	Seldom/never Several times a month Once a week Several times a week Every day Several times a day	Lower consumption (sum of <20) Higher consumption)sum of ≥20)
Dairy products (Total consumption indicated by the sum of all items)	“How often do you eat or drink any of the following foods or drinks, even in small quantities?” (1) Dairy, (2) fermented dairy products:	
	Seldom/never Several times a month Once a week Several times a week Every day Several times a day	Lower consumption (sum of <7) Higher consumption (sum of ≥7)
Different Oils (Total consumption indicated by the sum of all items)	“How often do you eat any of the following foods, even in small quantities?” (1) Linseed oil, (2) refined oils, (3) olive oil, (4) cold-pressed oil:	
	Seldom/never Several times a month Once a week Several times a week Every day Several times a day	Lower consumption (sum of <9) Higher consumption (sum of ≥9)
Vegetables and/or Fruits (Total consumption indicated by the sum of all items)	“How often do you eat any of the following foods, even in small quantities?” (1) Vegetables, (2) fruits:	
	Seldom/never Several times a month Once a week Several times a week Every day Several times a day	Lower consumption (sum of <9) Higher consumption (sum of ≥9)
Supplements (Total consumption indicated by the sum of all items)	“How often do you use any of the following supplements?” (1) Fish oil supplements, (2) probiotic supplements, (3) omega 3 supplements:	
	Seldom/never Several times a month Once a week Several times a week Every day Several times a day	Lower consumption (sum of <4) Higher consumption (sum of ≥4)

**Table 2 nutrients-13-04235-t002:** Comparison of participants’ background characteristics between the participants stratified by the absence/presence of xerostomia.

Background Characteristics	STUDY GROUPS		Significance
	Non-Xerostomia (%)	Xerostomia N (%)	
Sex			
Males	439 (34.0)	26 (23.2)	0.021 ^^^
Females	854 (66.0)	86 (76.8)	
Age (in full years)			
Mean (sd)	54.4 (11.8)	60.5 (10.4)	<0.001 ^#^
Median (IQR)	55.0 (21)	63.0 (15)	
Residence			
Urban	917 (70.9)	89 (79.5)	
Peri-urban	227 (17.6)	11 (9.8)	0.093 ^^^
Rural	149 (11.5)	12 (10.7)	
Education (in years)			
Mean (SD)	14.8 (3.3)	14.0 (3.0)	
Median (IQR)	15.0 (5)	14.0 (4)	0.032 ^#^

^^^ Chi-square test (categorical variables); ^#^ Mann–Whitney U test.

**Table 3 nutrients-13-04235-t003:** Comparison of consumption of selected diet items between the participants stratified by the absence/presence of xerostomia.

Consumption of Diet Items	STUDY GROUPS #		
	No-Xerostomia N (% of Total)	Xerostomia N (% of Total)	Significance ^
CARBOHYDRATES			
Bread			
Low frequency	166 (13.8)	24 (22.9)	0.029
Medium frequency	251 (20.9)	16 (15.2)	
High frequency	785 (65.3)	65 (61.9)	
Rice and pasta			
Low frequency	699 (60.0)	58 (58.0)	
Medium frequency	363 (31.2)	28 (28.0)	0.222
High frequency	103 (8.8)	14 (14.0)	
Cereals			
Low frequency	517 (45.2)	46 (46.0)	
Medium frequency	417 (36.5)	35 (35.0)	0.957
High frequency	210 (18.4)	19 (19.0)	
Potatoes			
Low frequency	334 (27.6)	31 (29.8)	
Medium frequency	534 (44.2)	41 (39.4)	0.645
High frequency	341 (28.2)	32 (30.8)	
PROTEINS			
Meat			
Low frequency	188 (15.6)	23 (21.5)	
Medium frequency	489 (40.5)	34 (31.8)	0.122
High frequency	530 (43.9)	50 (46.7)	
Processed meat products			
Low frequency	509 (49.0)	58 (64.4)	
Medium frequency	312 (30.0)	17 (18.9)	0.016
High frequency	218 (21.0)	15 (16.7)	
Fat fish			
Low frequency	795 (73.5)	55 (59.2)	0.009
Medium frequency	220 (20.4)	31 (33.3)	
High frequency	66 (6.1)	7 (7.5)
Lean fish			
Low frequency	871 (75.2)	63 (61.8)	0.009
Medium frequency	226 (19.5)	29 (28.4)	
High frequency	61 (5.3)	10 (9.8)	
Eggs			
Low frequency	413 (34.4)	37 (35.6)	
Medium frequency	520 (43.3)	44 (42.3)	0.970
High frequency	267 (22.3)	23 (22.1)	
Nuts and seeds			
Low frequency	693 (60.3)	58 (57.5)	
Medium frequency	281 (24.5)	27 (26.7)	0.841
High frequency	175 (15.2)	16 (15.8)	
OILS			
Linseed oil			
Low frequency	919 (84.3)	81 (81.8)	
Medium frequency	95 (8.7)	8 (8.1)	0.512
High frequency	76 (7.0)	10 (10.1)	
Olive oil			
Low frequency	659 (60.6)	56 (57.7)	
Medium frequency	287 (26.3)	26 (26.8)	0.773
High frequency	142 (13.1)	15 (15.5)	
Refined oils			
Low frequency	500 (46.2)	48 (49.5)	
Medium frequency	345 (31.9)	28 (28.9)	0.792
High frequency	238 (22.0)	21 (21.6)	
Cold-pressed oils			
Low frequency	821 (81.5)	72 (80.9)	0.013
Medium frequency	130 (12.9)	6 (6.7)	
High frequency	56 (5.6)	11 (12.4)	
VEGETABLES and FRUITS			
Vegetables			
Low frequency	101 (8.3)	8 (7.5)	0.792
Medium frequency	253 (21.0)	18 (16.8)	
High frequency	853 (70.7)	81 (75.7)	
Fruits			
Low frequency	238 (19.4)	22 (20.0)	
Medium frequency	380 (31.0)	37 (33.6)	0.792
High frequency	609 (49.6)	51 (46.4)	
DAIRY PRODUCTS			
Dairy (non-fermented)			
Low frequency	610 (52.5)	268 (23.1)	
Medium frequency	268 (23.1)	17 (16.3)	0.288
High frequency	284 (24.4)	27 (26.0)	
Fermented dairy products			
Low frequency	544 (46.5)	41 (39.8)	
Medium frequency	377 (32.2)	42 (40.8)	0.205
High frequency	248 (21.3)	20 (19.4)	
DESSERTS WITH ADDED SUGAR			
Low frequency	360 (36.0)	30 (36.1)	0.848
Medium frequency	304 (30.4)	23 (27.7)	
High frequency	336 (33.6)	30 (36.1)	
SWEETENED DRINKS			
Low frequency	327 (33.5)	31 (35.6)	
Medium frequency	345 (35.3)	33 (37.9)	0.651
High frequency	305 (31.2)	23 (26.4)	
DIFFERENT SUPPLEMENTS			
Fish oil supplements			
Low frequency	883 (80.0)	79 (79.8)	
Medium frequency	73 (6.6)	6 (6.1)	0.961
High frequency	148 (13.4)	14 (14.1)	
Probiotic supplements			
Low frequency	988 (93.1)	77 (83.7)	
Medium frequency	34 (3.2)	9 (9.8)	0.002
High frequency	39 (3.7)	6 (6.5)	
Omega 3 supplements			
Low frequency	869 (77.5)	79 (80.6)	
Medium frequency	65 (5.8)	7 (7.2)	0.479
High frequency	187 (16.7)	12 (12.1)	

^ Chi-square test. # Different numbers in comparison groups due to missing data.

**Table 4 nutrients-13-04235-t004:** Association between major diet groups and xerostomia adjusted with sociodemographic characteristics in adults.

Determinants	Single Determinant Models Crude ORs # (95% CI)	Final Multivariable Model ^ (after 7 Steps) Model Summary: Nagelkarke R^2^ = 0.739; *p* < 0.001. Determinants: Adjusted ORs (95% CI)
Sociodemographic characteristics		
Sex (males vs. females)	0.25 (0.22; 0.28)	NS
Age (55+ yrs vs. < 55 yrs)	0.23 (0.20; 0.26)	1.66 (1.03; 2.69)
Diet groups (consumption: higher vs. lower)		
Carbohydrates	0.23 (0.20; 0.26)	0.39 (0.23; 0.65)
Proteins	0.20 (0.17; 0.24)	0.56 (0.32; 0.99)
Oils	0.19 (0.16; 0.22)	0.58 (0.34; 1.00)
Fruits and vegetables	0.25 (0.22; 0.28)	NS
Dairy	0.21 (0.18; 0.24)	NS
Sweetened drinks	0.20 (0.17; 0.23)	NS
Desserts with added sugar	0.20 (0.17; 0.23)	NS
Supplements	0.21 (0.18: 0.24)	NS

# Crude ORs calculated from un-adjusted (univariable) binary logistic regression models. ^ Multivariable binary logistic regression, testing of determinants with Backward LR. NS—not statistically significant.

## Data Availability

The dataset analyzed during the current study is available from the corresponding author upon reasonable request.

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
