# Peer review of "Association between Diet and Xerostomia: Is Xerostomia a Barrier to a Healthy Eating Pattern?"

_nutrients, 2021, doi:10.3390/nu13124235_

Round 1
Reviewer 1 Report
The authors present the results of a cross-sectional study which was aimed to examine the association between xerostomia and diet among the Lithuanian population.
Overall, the work is interesting and the results are clearly presented. Study design was adequate and data handling and analysis were appropriate. The conclusions are well supported by the results.
However I should point some minor issues that, if corrected, might improve the scientific soundness of the manuscript.
Xerostomia evaluation criteria is based on a self-reported information gathered from a four category answer ('always', 'often', 'sometimes', 'never'). It would be interesting to report the frequency for each of the categories.
The multivariate binary logistic regression analysis procedure that was used is not clear. A model is presented but it is not stated the model performance iteration neither which was the variable inclusion procedure (stepwise?). How was the criteria for variable (sociodemographic and diet items) inclusion in the model? Is the presented model the best reduced model that is possible to achieve?
When accounting for multiple variables in a logistic regression, their specific impact in the model varies when adjusted, that's clear for everyone, but the model is also sensitive to the number of included variables.
Besides this it is also important is to establish clear criteria to handle variable selection. The differences on frequency consumption of 'cold pressed oil', 'bread', 'processed meat products', 'fat fish', 'lean fish' and 'probiotic supplements' were identified as statistically significant when comparing the non-xerostomia vs. xerostomia group. But any other diet item/variable could be accommodated in the final model? Sometimes variables that individually are not identified as statistically significant emerge as relevant in a multivariate model. I would suggest a stepwise logistic regression approach and also the presentation of crude OR for each variable (Table 3), prior to their inclusion in the model.
Reviewer 2 Report
This manuscript is a nice overview about the association between diet and xerostomia. It provides first insights about the relationship between xerostomia and eating patterns in patients with autoimmune diseases.
Remarks to the authors:
- chapter 2.2 how were the participants for the questionnaire chosen? Was there any preselection of potential candidates? Was there a specific access of patients with autoimmune diseases or xerostomia?
- Chapter3: Dryness of the mouth or xerostomia is a common symptom of menopausal women. Thus, "the higher proportion of participants reporting xerostomia were females, older age ...." might be biased, if not strictly asked or corrected for in the analysis. The number of children being born per women hasn't been considered either.
- Overall, the usage of the term "odds" seems somehow inappropriate. I would prefer the term "risk".
Round 2
Reviewer 2 Report
The authors have responded to my concerns in an appropriate way. I have no further comments. However, there are quite some spelling errors due to line-breaks such as:
lo-gistic on page 1, dis-eases on page 2, of-ten on page 2 .....
fre-quency on page 5.....
......
consump-tion and syn-drome on page 11